# Interleukin-34 Enhances the Tumor Promoting Function of Colorectal Cancer-Associated Fibroblasts

**DOI:** 10.3390/cancers12123537

**Published:** 2020-11-27

**Authors:** Eleonora Franzè, Antonio Di Grazia, Giuseppe Sigismondo Sica, Livia Biancone, Federica Laudisi, Giovanni Monteleone

**Affiliations:** 1Department of Systems Medicine, University of Rome “TOR VERGATA”, 00133 Rome, Italy; frnlnr01@uniroma2.it (E.F.); adigrazia2000@yahoo.it (A.D.G.); biancone@med.uniroma2.it (L.B.); federica.laudisi@uniroma2.it (F.L.); 2Department of Surgery, University “TOR VERGATA” of Rome, 00133 Rome, Italy; Giuseppe.Sica@uniroma2.it

**Keywords:** colon cancer, stromal cells, netrin-1, b-FGF, cytokines

## Abstract

**Simple Summary:**

In colorectal cancer (CRC), cancer-associated fibroblasts (CAFs) promote tumor growth and progression through the synthesis of various molecules targeting the neoplastic cells. Here, we demonstrate that IL-34, a cytokine highly expressed in CRC tissue, regulates the function of CAFs in a paracrine and autocrine manner. Specifically, IL-34 induces normal fibroblasts (NFs) to acquire a cellular phenotype resembling that of CAFs, while IL-34 knockdown in CAFs reduces their tumorigenic properties and proliferation. Moreover, IL-34 stimulates NFs to produce netrin-1 and b-FGF—two factors that enhance CRC cell growth and migration. Altogether, our data support the involvement of IL-34 in CRC.

**Abstract:**

The stromal compartment of colorectal cancer (CRC) is marked by the presence of large numbers of fibroblasts, termed cancer-associated fibroblasts (CAFs), which promote CRC growth and progression through the synthesis of various molecules targeting the neoplastic cells. Interleukin (IL)-34, a cytokine over-produced by CRC cells, stimulates CRC cell growth. Since IL-34 also regulates the function of inflammatory fibroblasts, we hypothesized that it could regulate the tumor promoting function of colorectal CAFs. By immunostaining and real-time PCR, we initially showed that IL-34 was highly produced by CAFs and to lesser extent by normal fibroblasts isolated from non-tumoral colonic mucosa of CRC patients. CAFs and normal fibroblasts expressed the functional receptors of IL-34. IL-34 induced normal fibroblasts to express α-SMA, vimentin and fibroblast activation protein and enhanced fibroblast growth, thus generating a cellular phenotype resembling that of CAFs. Consistently, knockdown of IL-34 in CAFs with an antisense oligonucleotide (AS) decreased expression of such markers and inhibited cell proliferation. Co-culture of CRC cells with IL-34 AS-treated CAFs supernatants resulted in less cancer cell proliferation and migration. Among CAF-derived molecules known to promote CRC cell growth/migration, only netrin-1 and basic-fibroblast growth factor were induced by IL-34. Data suggest a role for IL-34 in the control of colorectal CAF function.

## 1. Introduction

In carcinomas, neoplastic epithelial cells actively interact with various immune cells, stromal cells and endothelial cells, thus generating a microenvironment that fosters carcinogenetic processes. Specifically, the tumor-associated stromal cells play a pivotal role in tumor initiation, progression, drug resistance, and relapse of cancer [1]. This has been shown, for instance, in colorectal cancer (CRC)—one of the most common cancers accounting for nearly 10% of cancer-related deaths worldwide [2]. Large numbers of fibroblasts, which express α-smooth muscle actin (α-SMA), are evident in the stromal compartment of CRC [3]. Functional studies showed that such cells, termed cancer-associated fibroblasts (CAFs), promote malignant cell growth and metastasis through bidirectional signaling with both neoplastic cells and other cells within the tumor microenvironment [4,5]. Moreover, specific stromal gene profiles have been associated with resistance to chemotherapy and poor prognosis [6,7] Within the tumor microenvironment, both immune and non-immune cells synthesize various cytokines, which trigger specific signaling in fibroblasts, thereby instructing them to produce a pro-tumorigenic secretome [8].

Interleukin-34 (IL-34), a cytokine produced by a variety of immune and non-immune cells, is spontaneously produced in the human gut, and its expression is up-regulated in the inflamed gut of patients with inflammatory bowel disease (IBD) and in CRC tissue [9,10,11,12]. Specifically, both CRC cells and non-tumoral cells infiltrating CRC tissue express IL-34, and the cytokine stimulates CRC cell growth [9]. Moreover, by using fibroblasts isolated from IBD patients, we have recently shown that IL-34 induces collagen synthesis [13]. Altogether, these observations raise the possibility that IL-34 can mediate the interplay between cancer cells and stromal cells during the colon carcinogenesis process.

In the present study, we sought to ascertain whether IL-34 regulates the CAF-mediated tumor-promoting functions.

## 2. Results

### 2.1. Intestinal Fibroblasts Express Interleukin-34 and Its Receptors

We extracted fibroblasts from surgical specimens of both tumor tissue and normal, adjacent colonic mucosa (at least 2 cm from the outer tumor margin) of patients undergoing resection for CRC. The tumor masses were dissociated, and various cell types were separated to obtain populations of CAFs and normal fibroblasts. The experiments were performed by comparing these pairs of CAFs and the corresponding normal fibroblasts, thereby avoiding bias due to inter-individual differences. Initially, we verified the purity of the various fibroblast populations by immunostaining. These fibroblast populations strongly expressed the fibroblastic marker vimentin and were negative for cytokeratin and CD31 (Figure 1A), thus indicating that our fibroblast populations were prepared with minimal contamination by epithelial cell and endothelial cells. Flow cytometry analysis confirmed that both CAFs and NFs expressed α-SMA, while they were negative for CD31 (endothelial marker) and EpCAM (epithelial marker) (Appendix A).

By real-time PCR, we next showed that IL-34 RNA transcripts were significantly increased in CAFs compared to the corresponding normal fibroblasts, and this result was confirmed by immunostaining (Figure 1B,C). Moreover, IL-34 expression was more pronounced in fibroblasts isolated from ulcerative colitis-associated cancer as compared to normal fibroblasts (Figure 1C). Flow cytometry analysis confirmed the major expression of IL-34 in CAFs as compared to NFs (Appendix A).

Real-time PCR and immunostaining analysis showed that CAFs and normal fibroblasts expressed both M-CSFR-1 and PTP-ζ, two functional receptors of IL-34 [14,15] (Figure 2).

### 2.2. Interleukin-34 Induces an Activated Phenotype in Intestinal Fibroblasts

Comparative analysis of gene-expression profiles of CAFs and fibroblasts isolated from non-tumoral tissues showed that fibroblasts in tumor masses possess biological characteristics distinct from those of normal fibroblasts [16,17]. To assess whether IL-34 induces a cellular phenotype resembling that of CAFs, we initially assessed whether IL-34 induces in normal fibroblasts markers, which are over-expressed by CAFs [16]. RNA expression of α-SMA, vimentin and fibroblast activation protein (FAP) was increased in IL-34-treated normal fibroblasts as compared to untreated cells (Figure 3A–C), while RNA transcript of each molecule was decreased in CAFs following IL-34 knockdown with an antisense oligonucleotide (AS) (Figure 3D–G). It has long been recognized that CAFs proliferate in vitro at much higher rates than activated fibroblasts isolated from normal tissue or from tissue with acute injury or healing wounds [18].

Therefore, we cultured normal intestinal fibroblasts with IL-34 and then evaluated the rate of proliferation (Figure 4A). Basic-fibroblast growth factor (b-FGF) was used as a positive control [19,20]. Flow-cytometry analysis of CSFE-stained cells and BrdU analysis showed that IL-34 significantly enhanced proliferation of normal fibroblasts (Figure 4A, Appendix A). Consistently, inhibition of IL-34 with the specific AS decreased CAFs proliferation (Figure 4B, Appendix A).

### 2.3. Interleukin-34 Stimulates Cancer-Associated Fibroblasts to Promote Colon Cancer Cell Growth and Migration

In co-culture experiments, CAFs enhance tumorigenesis of cancer cells when compared with normal fibroblasts [21]. Therefore, we assessed whether IL-34 produced by CAFs could enhance CRC cell growth and migration. To this end, we cultured DLD-1 cells in the presence or absence of IL-34 AS-treated CAFs-conditioned media and then examined the proliferation of these cells. DLD-1 were selected because these cells are known to proliferate vigorously in response to CAF-derived factors [22]. DLD-1 cells cultured in the presence of IL-34 AS-treated CAFs supernatants exhibited a significant reduction in proliferation as compared to DLD-1 cells cultured in the presence of supernatants of CAFs treated with the control AS (Figure 5A). Moreover, a scratch test showed that the supernatants of IL-34-deficient CAFs had reduced ability to promote DLD-1 cell migration as compared to the supernatants of untreated CAFs (Figure 5B). Similar results were seen when experiments were performed using HT-29 cells (Appendix A).

### 2.4. Interleukin-34 Enhances Netrin-1 and b-FGF Expression in CAFs

The above data prompted us to test the possibility that IL-34 could stimulate CAFs to produce factors promoting CRC cell growth and migration. To this end, we assessed the RNA transcripts for several molecules, which are known to stimulate CRC cell proliferation/migration. Among these factors, only netrin-1 and b-FGF RNA expression was significantly increased in normal fibroblasts stimulated with IL-34 (Figure 6A). In line with such results, knockdown of IL-34 with the AS decreased netrin-1 and b-FGF RNA transcripts in CAFs (Figure 6B).

As expected, netrin-1 and b-FGF protein content was increased in normal fibroblasts treated with IL-34 (Figure 7A,B), while such proteins were down-regulated in IL-34 AS-treated CAFs (Figure 7C,D).

## 3. Discussion

This study was undertaken to examine whether IL-34, a cytokine highly expressed in CRC tissue, regulates the function of CAFs. Indeed, it has long been recognized that cytokines produced within the tumor environment contribute to the colon carcinogenesis process by targeting CAFs and, hence, regulating many aspects of tumor progression, from in situ growth of primary tumors to metastatic spread of cancer cells [8]. Data of the present study show that CAFs produce IL-34 and express the receptors for this cytokine. These data, together with our previous results showing that CRC cells can produce IL-34 [9,23], raise the possibility that IL-34 can regulate CAF functions in a paracrine and autocrine manner. Normal stroma in most organs contains a small number of quiescent or resting fibroblasts, whereas reactive tumor stroma is generally characterized by an increased number of activated fibroblasts that express αSMA, vimentin and FAP [17]. Notably, IL-34 enhanced expression of such markers in normal intestinal fibroblasts, while inhibition of IL-34 expression in CAFs decreased αSMA, vimentin and FAP. It is, however, known that such markers are not sufficient to identify CAFs with pro-tumorigenic activity, and some CAF markers may also be associated with cells with diverse, and possibly opposing, functions in the context of the specific tumor microenvironment [24]. Therefore, to further support the ability of IL-34 to promote differentiation of CAFs with tumorigenic properties, we performed functional studies and evaluated the effect of IL-34 stimulation on CAF proliferation. Indeed, it is known that CAFs proliferate vigorously and such a proliferation is regulated by cytokines released within the tumor microenvironment [21]. Flow-cytometry analysis revealed that IL-34 stimulated the growth of normal fibroblasts as well as inhibition of IL-34 was accompanied by reduced proliferation of CAFs. In recent years, evidence has been accumulated to show that in co-culture experiments, CAFs enhance tumorigenesis of cancer cells when compared with normal fibroblasts [19]. Pioneering studies showed that CAFs but not normal fibroblasts promoted the formation of tumors resembling prostatic cancer when inoculated into mice with Simian virus 40-transformed prostate epithelial cells [25]. Subsequently, similar results were obtained in other cancer systems where the ability of CAFs to influence tumor growth was partly dependent on the function of CAF-derived cytokines [26,27]. Our results fit with such observations and highlight the proliferative effect of IL-34 on cancer cells, as knockdown of IL-34 in CAFs attenuated the ability of culture supernatants of such cells to promote DLD-1 cell growth. Moreover, supernatants of IL-34-depleted CAFs exhibited reduced ability to promote in vitro DLD-1 cell migration. These findings confirm and expand on results of our previous studies showing a direct mitogenic effect of IL-34 on CRC cells [9,23]. Next, we explored the possibility that IL-34 can stimulate CAFs to produce factors that regulate CRC cell behavior. To this end, we analyzed the expression of several factors involved in the regulation of CRC cell growth and migration in normal fibroblasts stimulated with IL-34. IL-34 enhanced expression of netrin-1 and b-FGF, and this effect was evident at both RNA and protein level. Consistently, both netrin-1 and b-FGF were down-regulated in IL-34-depleted CAFs. Netrin-1 is a multifunctional secreted glycoprotein involved in the control of several biological processes, such as angiogenesis, neuronal navigation, angiogenesis, and cell survival and migration and accumulating evidence suggests a role for this protein in many pathologies, including cardiovascular diseases, diabetes, and cancer [28,29,30]. Netrin-1 is highly expressed in CAFs in CRC tissue and regulates CRC cell stemness [31,32]. Similarly b-FGF stimulates acquisition of CAFs metastatic capacity [33] and positively regulates the growth of CRC cells [34]. The factors that control Netrin-1 and b-FGF induction in CRC remain, however, poorly characterized. Data of the present study indicate that IL-34 is a regulator of Netrin-1 and b-FGF in colon CAFs and suggest that the IL-34 promoting effect on CRC cell growth and migration can, at least in part, rely on the induction of these two proteins by CAFs. In conclusion, this study shows that CAFs produce and respond to IL-34 with the downstream

## 4. Materials and Methods

### 4.1. Patients and Samples

Paired tissue samples were taken from the tumoral area and the macroscopically and microscopically unaffected, adjacent colonic mucosa of 9 patients who underwent colon resection for sporadic CRC at the Tor Vergata University Hospital (Rome, Italy). All patients received neither radiotherapy nor chemotherapy prior to undergoing surgery. An additional surgical sample was taken from the tumoral area of 3 patients with colitis-associated colon cancer (CAC). Each patient who took part in the study gave written informed consent, and the study protocol was approved by the local Ethics Committee (Tor Vergata University Hospital, Rome, Italy; protocol number: 171/16, received in 19 December 2016).

### 4.2. Isolation and Culture of Intestinal Fibroblasts

All the reagents were purchased from Sigma-Aldrich (Milan, Italy) unless specified. Intestinal fibroblasts were isolated from the tumoral and non-tumoral area of 9 CRC patients as described elsewhere [35,36]. Fibroblasts were maintained in 75 cm^2^ plastic flasks and incubated at 37 °C in a humidified atmosphere with 5% CO_2_ in D-MEM containing high glucose with Ultra Glutamine and supplemented with 10% fetal bovine serum (FBS), 1% of penicillin (100 U/mL), streptomycin (100 μg/mL) and 1% of non-essential amino acids (all from Lonza, Verviers, Belgium). Depending on the studies, fibroblasts were used either freshly isolated (for evaluating either the cell phenotype or the content of the cytokine and its receptors) or after 3–8 passages.

To assess whether IL-34 controls cell proliferation, 5 × 10^4^ normal fibroblasts were plated into each well of a 12-well plate, left to adhere for 24 h, then starved for 6 h and finally either left unstimulated or stimulated with IL-34 (50 ng/mL, Miltenyi Biotec, Bologna, Italy) and b-FGF (20 ng/mL, Peprotech EC, Ltd., London, UK). After 48 h, cell proliferation was assessed by flow cytometry. In further experiments, normal fibroblasts were plated into each well of a 12-well plate, left to adhere for 24 h, then starved for 6 h and finally either left unstimulated or stimulated with recombinant human IL-34 for 6–48 h. At the end, cells were used for protein and gene expression analysis by Western blotting or Real time PCR, respectively, and cell-free supernatants were stored at −80 °C until used.

CAFs (5 × 10^4^) were plated and left to adhere as indicated above and after 24 h transfected with an IL-34 antisense oligonucleotide (IL-34 AS) or negative control antisense oligonucleotide (NCAS) (both used a 200 nM, Integrated DNA Technologies, Inc. Leuven, Belgium) for 24–48 h using Opti-MEM medium and Lipofectamine 3000 reagent according to the manufacturer’s instructions (both from Life Technologies, Milan, Italy). Cells were used for analysis of RNA, protein and proliferation. Cell-free supernatants were used for assessing their ability to induce DLD-1 cell proliferation.

### 4.3. Real-Time PCR

A constant amount of RNA (0.5 μg/sample) was retro-transcribed into complementary DNA (cDNA) and then 1 μL of cDNA/sample was amplified using the following conditions: denaturation 1 min at 95 °C; annealing 30 s, at 57 °C for: TGF-α, at 58 °C for: M-CSFR-1, PTP-ζ, IGF2, FAP, Vimentin; at 60 °C for: β-Actin, IL-34, α-SMA, IGF1; HGF, b-FGF, and at 61 °C for Netrin-1, followed by 30 s of extension at 72 °C. The following primer sequences were used: IL-34: forward, 5′-ACAGGAGCCGACTTCAGTAC-3′ and reverse, 5′-ACCAAGACCCACAGATACCG-3′; M-CSFR-1: forward, 5′-CTGCTCAACTTTCTGCGAAG-3′ and reverse, 5′-CTCATCTCCACATAGGTGTC-3′; PTP-ζ: forward, 5′-ACTAACCGATCCCCAACAAG-3′ and reverse, 5′-CCACACATTTCCCTCCATAG-3′; α-SMA: forward 5′-TCTGGAGATGGTGTCACCCA-3′ and reverse 3′-ACCCACTGTGGTAGAGGTCT-5′; FAP: forward, 5′-TGTCCACCTTCCACAAGTAC-3′ and reverse, 5′-CTGTCCAAGTTGCTCATCAG-3′; Vimentin: forward, 5′-GGAGCAGCAGAATAAGATCC-3′ and reverse, 5′-CCAGAGACGCATTGTCAAC-3; IGF1: forward, 5′-CTCCTCGCATCTCTTCTACC-3′ and reverse, 5′-CTCCAGCCTCCTTAGATCAC-3′; IGF2: forward, 5′-TCGTTGAGGAGTGCTGTTTC-3′ and reverse, 5′-GGGGTATCTGGGGAAGTTG-3′; TGF-α: forward, 5′-CTGTTCGCTCTGGGTATTG-3′ and reverse, 5′-ACTGAGTGTGGGAATCTGG-3′; HGF: forward, 5′-TCCAGAGGTACGCTACGAAG-3′ and reverse, 5′-GGTGTGGTGTCTGATGATCC-3′; b-FGF: forward, 5′-AGAGCGACCCTCACATCAAG-3′ and reverse, 5′-AGCCAGGTAACGGTTAGCAC-3′; Netrin-1: forward, 5′-GAAGACTGCGATTCCTACTGC-3′ and reverse, 5′-CCCTGCTTATACACGGAGATG-3′; β-actin: forward, 5′-AAGATGACCCAGATCATGTTTGAGACC-3′ and reverse 5′-AGCCAGTCCAGACGCAGGAT-3′. mRNA expression was calculated relative to the housekeeping β-Actin gene on the basis of the ∆∆Ct algorithm.

### 4.4. Total Protein Extraction and Western Blotting

Fibroblasts were lysed on ice in buffer containing 10 mM HEPES (pH 7.9), 10 mM KCl, 0.1 mM EDTA, 0.2 mM EGTA and 0.5% Nonidet P40 supplemented with 1 mM dithiothreitol, 10 mg/mL aprotinin, 10 mg/mL leupeptin, 1 mM phenylmethylsulfonyl fluoride, 1 mM Na_3_VO_4_ and 1 mM NaF. Lysates were clarified by centrifugation at 4 °C for 30 min, and separated on 10% sodium dodecyl sulphate-polyacrylamide gel electrophoresis, and membranes were then incubated with the following antibodies: mouse anti human IL-34 (1:1000 Abcam, Cambridge, UK), rabbit anti-human Netrin-1 (1:1000 Abcam), rabbit anti-human b-FGF (final dilution 1:1000, Abcam), followed by horseradish peroxidase–conjugated secondary IgG monoclonal antibodies (all used at final dilution 1:20,000, Dako, Milan, Italy). The reaction was detected with a sensitive enhanced chemiluminescence kit (Pierce, Rockford, IL, USA). After the analysis, blots were stripped and incubated with the following internal loading control: mouse anti-human β-Actin (final dilution 1:5000 Sigma-Aldrich). Computer-assisted scanning densitometry (Image-Lab 5.2.1, Bio-Rad Laboratories, Milan, Italy) was used to analyze the intensity of the immunoreactive bands.

### 4.5. Immunofluorescence

CAFs and normal fibroblasts were assessed by immunofluorescence as indicated elsewhere [13] using the following antibodies: mouse anti-human IL-34 (final dilution 1:200, Abcam), mouse anti-human Vimentin (final dilution 1:100, ThermoFisher Scientific, Milan, Italy), mouse anti-human CD31 (final dilution 1:100, ThermoFisher Scientific), mouse anti-human Cytocheratin (final dilution 1:100, Santa Cruz Biotechnology, Inc., Dallas, TX, USA), mouse anti-human M-CSFR-1 (final dilution 1:100, Novus Biological, CO, USA) or mouse anti-human PTP-ζ (final dilution 1:100, Abcam). Slides were then incubated for 1 h at RT with specific secondary antibodies coupled with Alexa Fluor Dyes (final dilution 1:2000; ThermoFisher Scientific). Coverslips were mounted on glass slides using ProLong Gold antifade reagent with DAPI (ThermoFisher Scientific) to counterstain the DNA. Samples were analyzed with a Leica DMI 4000 B fluorescence microscope (Leica, Wetzlar, Germany).

### 4.6. Flow Cytometry

Cells were stained with the following monoclonal anti-human antibodies: PerCP 5.5 anti-EpCAM (both from Becton Dickinson, Milan, Italy), Pacific Blue anti-CD31 (eBioscience, San Diego, CA, USA), fluorescein isothiocyanate (FITC) anti-SMA, phycoerythrin (PE) anti-IL-34 (both from R&D Systems, Minneapolis, MN, USA). In all experiments, appropriate isotype control IgG was used. All antibodies were used at 1:50 final dilution. For intracellular staining, cells were fixed and permeabilized using IC Fixation buffer and the permeabilization buffer (both from eBioscience) according to the manufacturer’s instruction. Cells were analyzed by flow cytometry (FACSVerse, Becton Dickinson) and data were analyzed by FlowJo ver. 10.7 (Becton Dickinson).

### 4.7. Analysis and Quantification of Cell Proliferation

Cell proliferation was assessed by flow cytometry using carboxyfluorescein diacetate succinimidyl ester (CFSE; Molecular Probes, Life Technologies), which covalently binds cell components to yield a fluorescence that is divided equally between daughter cells at each division. Briefly, starved DLD-1 cells or normal fibroblasts or CAFs were incubated with CFSE according to the manufacturer’s instructions. After 30 min, the medium was removed, and fresh medium was added for the indicated time points. At the end of the specific treatment, cells were collected, washed twice with PBS and then incubated with PI (5 mg/mL, Life Technologies) for 15 min at 4 °C in the dark. CFSE- and/or PI-positive cells were determined by flow cytometry (FACSVerse BD Biosciences, San Jose, CA, USA), and the data were analyzed using ModFit LT 5.0 (Verity Software House, Inc., Topsham, ME, USA) and expressed as relative proliferation index with respect to unstimulated conditions.

Cell growth was evaluated by using a commercially available 5-bromodeoxyuridine (BrdU) assay kit (Roche Diagnostics, Monza, Italy). Briefly, 3500 cells were cultured in 96-well microplates and allowed to adhere overnight. Five-bromodeoxyuridine was added to the cell cultures 6 h before the end of the experiments, and cell growth was evaluated by ELISA at 450 nm.

### 4.8. Enzyme-Linked Immunosorbent Assay

Human b-FGF was measured using a sensitive commercial ELISA kit (R&D Systems) according to the manufacturer’s instructions.

### 4.9. Wound Healing Scratch Assay

A confluent DLD-1 or HT-29 cell monolayer was artificially wounded by scratching with a 200 μL pipette tip (“pseudo” wound approximately 1 mm diameter). The wells were washed with PBS to remove debris, and then the cell-free supernatants of CAFs previously transfected with sense oligonucleotide or IL-34 AS for 24 h were added for further 24 h. Images were taken at time 0 and after 24 h, and the “pseudo” wound area was measured by Image-Lab software (Bio-Rad Laboratories). The wound healing ability of DLD-1 at 24 h was expressed as a percentage of “pseudo” wound area with respect to that at time 0.

### 4.10. Statistical Analysis

Differences between two groups were compared using the Student’s *t*-test. All the analyses were performed using Graph-Pad 6 software (San Diego, CA, USA).

## 5. Conclusions

This is the first study showing that, in colon cancer, IL-34 is produced by CAFs and targets these cell populations with the down-stream effect of enhancing the CAF pro-tumorigenic function.

## Figures and Tables

**Figure 1 cancers-12-03537-f001:**
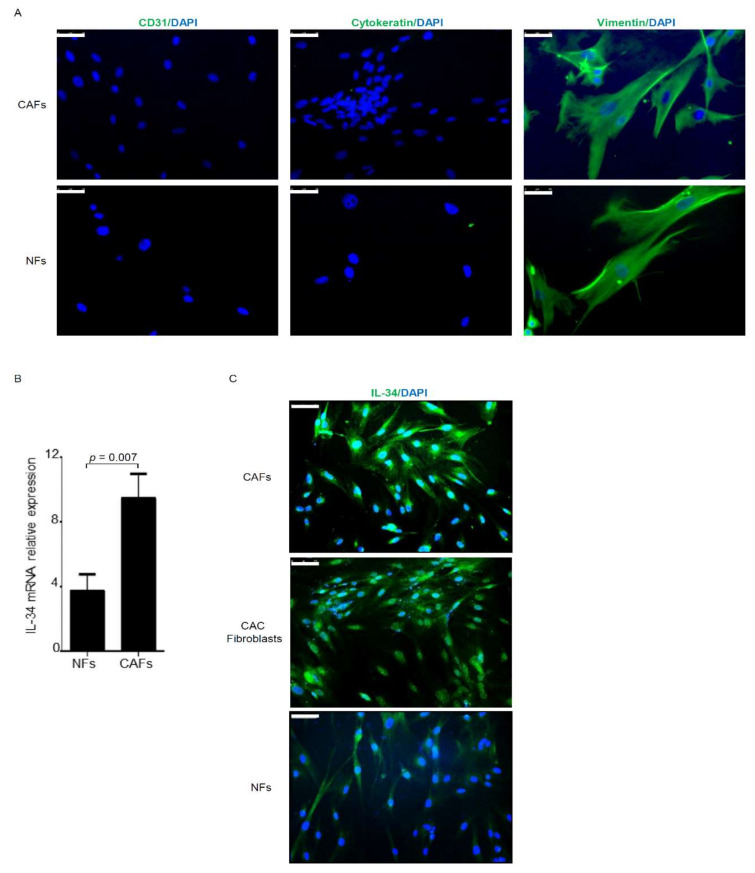
Cancer-associated fibroblasts express interleukin-34 (IL-34). (**A**) Representative images of immunofluorescence staining of fibroblasts isolated from tumoral (cancer-associated fibroblasts—CAFs) and non-tumoral (normal fibroblasts—NFs) area of one patient with colon cancer and analyzed for the expression of CD31 (green), cytocheratin (green), vimentin (green), and DAPI (blue). (Scale bars: 50 µm). One of three representative experiments, in which cells of three patients were used, is shown. (**B**) NFs and CAFs were evaluated for the expression of IL-34 by real-time PCR; levels were normalized to β-Actin and data were expressed as mean ± SEM of five patients. (**C**) Representative images of immunofluorescence staining of CAFs and NFs taken from tumoral and non-tumoral area, respectively, of one patient with colon cancer and one patient ulcerative colitis-associated colon cancer (CAC) and analyzed for the expression of IL-34 (green), and DAPI (blue). The example is representative of six separate experiments in which cells of three patients with colon cancer and three patients with CAC were analyzed. The scale bars are 75 µm.

**Figure 2 cancers-12-03537-f002:**
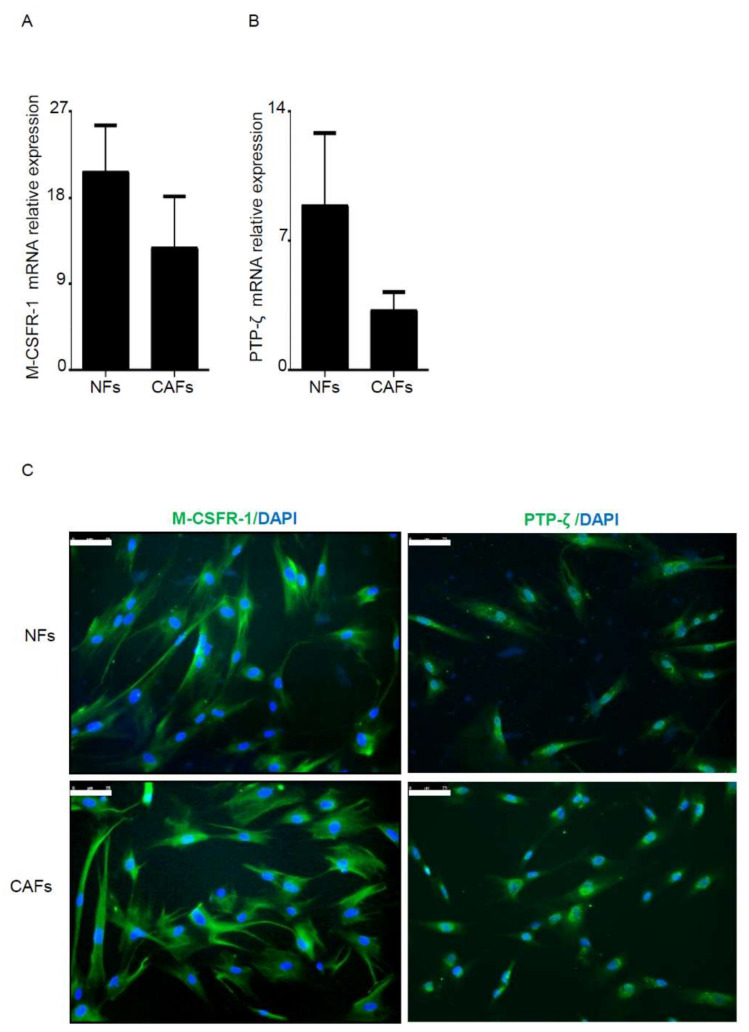
Both normal fibroblasts (NFs) and cancer-associated fibroblasts (CAFs) express IL-34 receptors. **A**–**B**: NFs and CAFs were evaluated for the expression of macrophage colony-stimulating factor receptor (M-CSFR-1) (**A**) and PTP-ζ (**B**) RNA transcripts by real-time PCR; levels were normalized to β-Actin and data were expressed as mean ± SEM of six patients. (**C**) Representative images of immunofluorescence staining of NFs and CAFs taken from tumoral and non-tumoral area, respectively, of one patient with colon cancer and analyzed for the expression of M-CSFR-1 (green), PTP-ζ (green), and DAPI (blue). The example is representative of three separate experiments in which cells of three patients with colon cancer were analyzed. The scale bars are 75 µm.

**Figure 3 cancers-12-03537-f003:**
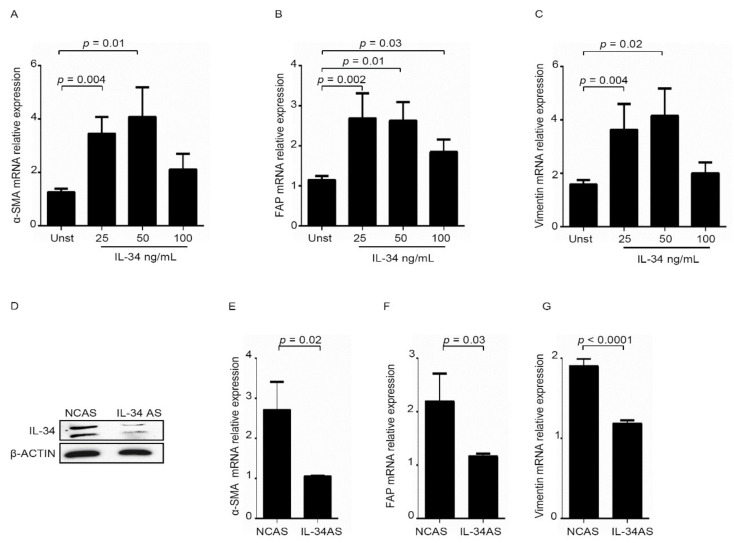
IL-34 stimulates normal fibroblasts (NFs) to acquire a tumoral phenotype. (**A**–**C**). Serum-starved NFs were either left unstimulated (UNST) or treated with IL-34 (25–100 ng/mL) for 6 h, and then α-SMA (**A**), FAP (**B**), Vimentin (**C**) RNA transcripts were evaluated by real-time PCR; levels were normalized to β-Actin and data were expressed as mean ± SEM of six experiments in which cells of six patients were analyzed. (**D**) CAFs were transfected with either negative control antisense oligonucleotide (NCAS) or IL-34 antisense oligonucleotide (IL-34 AS) for 48 h. IL-34 and β-Actin were then analyzed by Western blotting. Uncropped Western Blot images are available in Appendix A. (**E**–**G**). CAFs were transfected with negative control antisense oligonucleotide (NCAS) or IL-34 antisense oligonucleotide (IL-34 AS) for 24 h and α-SMA (**E**), FAP (**F**), Vimentin (**G**) RNA transcripts were analyzed by real-time PCR; levels were normalized to β-Actin, and data were expressed as mean ± SEM of six experiments in which cells of six patients were analyzed.

**Figure 4 cancers-12-03537-f004:**
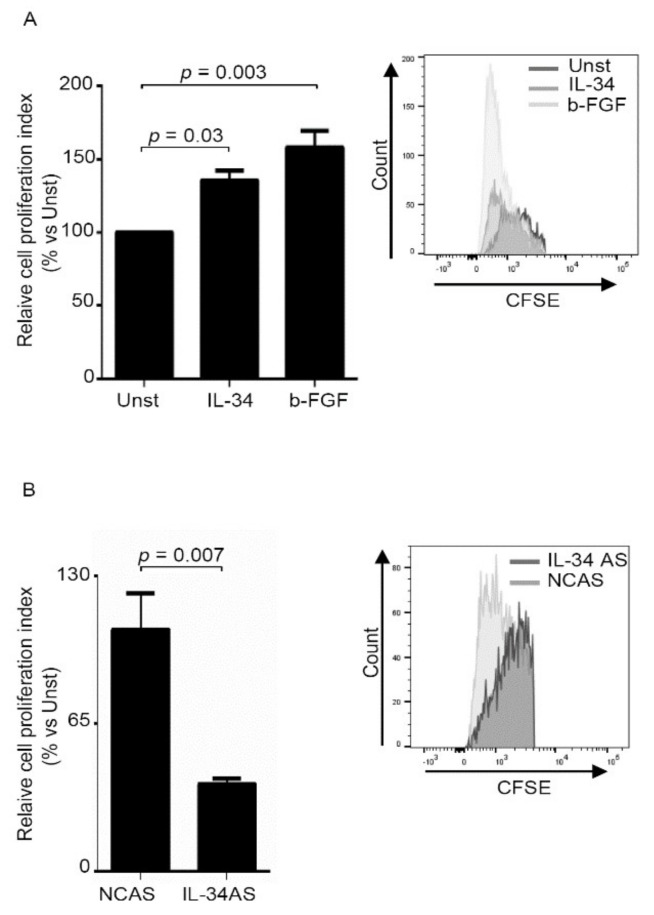
IL-34 enhances fibroblast proliferation. (**A**) Serum-starved normal fibroblasts (NFs) were stimulated with IL-34 (50 ng/mL) or basic-fibroblast growth factor (b-FGF) (20 ng/mL) for 48 h. Cell proliferation was evaluated by flow cytometry, and proliferation index was calculated with Modfit LT. Data indicate mean ± SEM of six independent experiments in which cells of six patients were analyzed. Right insets: representative histograms showing the expression of carboxyfluorescein diacetate succinimidyl ester (CFSE) in NFs analyzed by flow cytometry. (**B**) CAFs were transfected with negative control antisense oligonucleotide (NCAS) or IL-34 antisense oligonucleotide (IL-34 AS) for 48 h. Cell proliferation was evaluated by flow cytometry, and proliferation index was calculated with Modfit LT. Data indicate mean ± SEM of six independent experiments in which cells of six patients were analyzed. Right insets: representative histograms showing the expression of CFSE in CAFs analyzed by flow cytometry. UNST= unstimulated.

**Figure 5 cancers-12-03537-f005:**
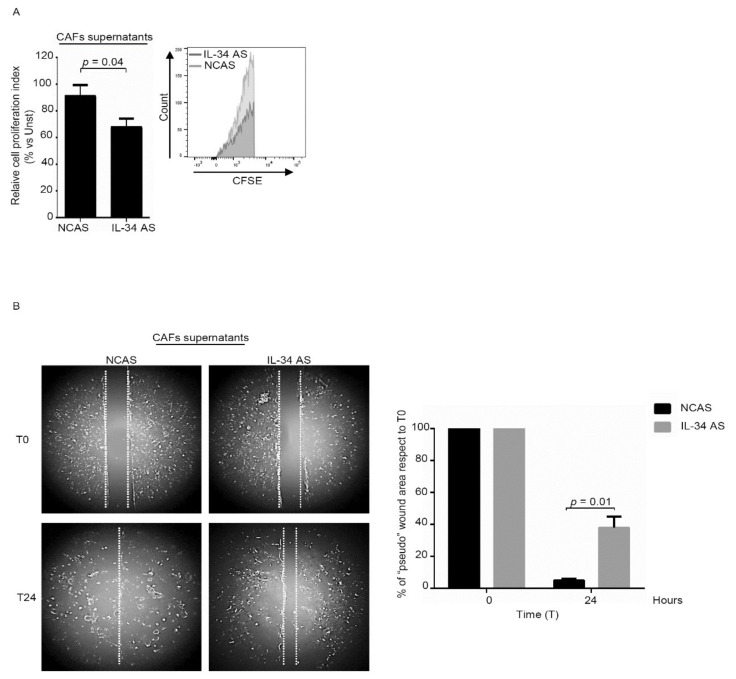
IL-34 produced by fibroblasts decreases DLD-1 cell proliferation and migration. (**A**) DLD-1 cells were incubated with supernatants of CAFs previously transfected with either negative control antisense oligonucleotide (NCAS) or IL-34 antisense oligonucleotide (IL-34 AS) for 24 h. DLD-1 proliferation was evaluated after 48 h by flow cytometry and proliferation index was calculated with Modfit LT. Data indicate mean ± SEM of six independent experiments in which cells of six patients were analyzed. Right insets: representative histograms showing the expression of CFSE in DLD-1 analyzed by flow cytometry. (**B**) Representative images of “pseudo” wound in monolayer of DLD-1 treated with supernatants of CAFs previously transfected with either negative control antisense oligonucleotide (NCAS) or IL-34 antisense oligonucleotide (IL-34 AS) for 24 h. Cells were photographed at the time of scratch (T0) and examined for cell migration after 24 h from the specific stimulation. Right panel shows the % of “pseudo” wound area in a monolayer of DLD-1 cells at the specific time point (T) with respect to area in T0 (defined as 100%), and the data are expressed as mean ± SEM of six independent experiments in which cells of six patients were analyzed.

**Figure 6 cancers-12-03537-f006:**
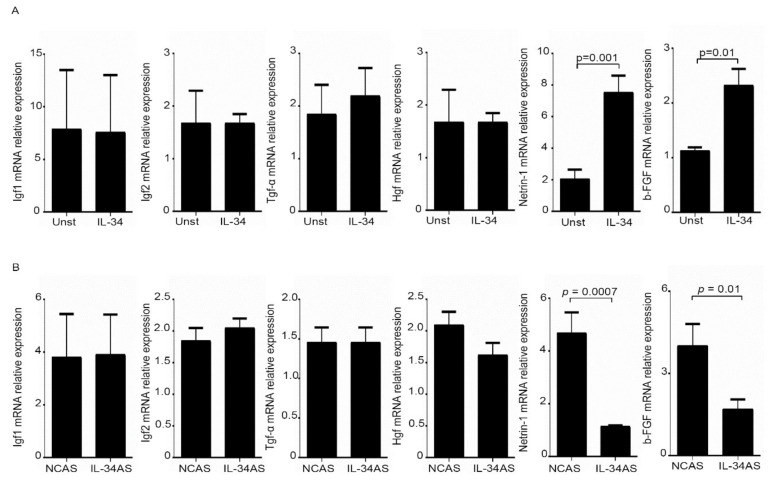
IL-34 induces Netrin-1 and b-FGF RNA expression. (**A**) Serum-starved normal fibroblasts (NFs) were treated with IL-34 (50 ng/mL) for 6 h and Igf1; Igf2, Tgf-α, Hgf, Netrin-1 and b-FGF RNA transcripts were analyzed by real-time PCR; levels were normalized to β-Actin and data were expressed as mean ± SEM of six experiments in which cells of six patients were analyzed. (**B**) CAFs were transfected with either negative control antisense oligonucleotide (NCAS) or IL-34 antisense oligonucleotide (IL-34 AS) for 24 h and Igf1; Igf2, Tgf-α, Hgf, Netrin-1 and b-FGF RNA transcripts were analyzed by real-time PCR; levels were normalized to β-Actin and data were expressed as mean ± SEM of six experiments in which cells of six patients were analyzed. UNST = unstimulated.

**Figure 7 cancers-12-03537-f007:**
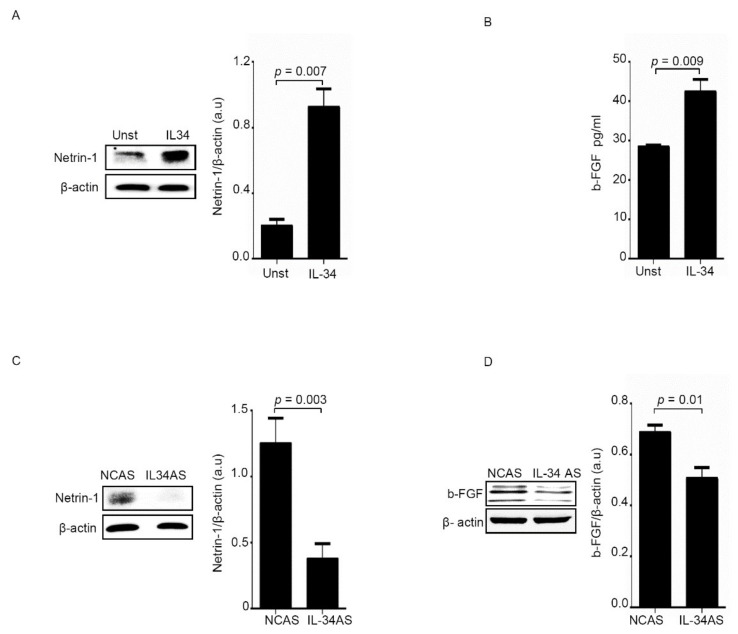
IL-34 induces Netrin-1 and b-FGF protein expression. (**A**) Serum-starved normal fibroblasts (NFs) were treated with IL-34 (50 ng/mL) for 24 h. Netrin-1 and β-Actin were evaluated by Western blotting. One of three independent experiments, in which cells of three patients were analyzed, is shown. Right panel shows the quantitative analysis of Netrin-1/β-Actin ratio. Values are expressed in arbitrary units (a.u.) and indicate mean ± SEM. Uncropped Western Blots images are available in Appendix A. (**B**) Serum-starved NFs were treated with IL-34 (50 ng/mL) for 48 h and b-FGF was evaluated in cell-free supernatants by ELISA. Values are expressed in arbitrary units (a.u.) and indicate mean ± SEM of six experiments in which cells of six patients were analyzed. C-D. CAFs were transfected with either negative control antisense oligonucleotide (NCAS) or IL-34 antisense oligonucleotide (IL-34 AS) for 24 h, and Netrin-1 and β-Actin (**C**) or b-FGF and β-Actin (Uncropped Western Blots images are available in Appendix A) (**D**) were evaluated by Western blotting. Right panels show the quantitative analysis of Netrin-1/β-Actin (**C**) or b-FGF /β-Actin (**D**) ratio, and values are expressed in arbitrary units (a.u.) and indicate mean ± SEM (Uncropped Western Blots images are available in Appendix A). One of three independent experiments, in which cells of three patients were analyzed, is shown. UNST = unstimulated.

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
