# Peer review of "Interleukin-34 Enhances the Tumor Promoting Function of Colorectal Cancer-Associated Fibroblasts"

_cancers, 2020, doi:10.3390/cancers12123537_

Round 1
Reviewer 1 Report
This paper investigates the role of IL-34 in colorectal cancer progression by analyzing cell proliferation and migration of CRC cells in response to IL-34 produced by tumor-associated fibroblasts from clinical specimens of IBD and CRC tissues. The manuscript demonstrates IL-34 is expressed by cultured cancer associated fibroblast (CAFs). The manuscript demonstrated the IL-34 is expressed by CAFs and CAC fibroblasts in figure 1C. However, it would have been preferable to conduct flow cytometry on primary cells isolated from IBD or CRC, fluorescently stained for CD31, cytokeratin, vimentin and intracellular IL-34. This would have better illustrated the population(s) of primary cells isolated from IBD or CRC. Cell proliferation was analyzed using flow cytometry in figure 4. However, another method to test cell proliferation, e.g., MTT assay, could have been used to use multiple methods to demonstrate IL-34's function. The manuscript should better describe and justify the use of the DLD-1 cell line. Overall, the manuscript demonstrated IL-34 produced by CAFs and CAC fibroblast cells can promote growth and migration of the DLD-1 CRC cell line.
Author Response
We thank the reviewer for his/her helpful suggestions
1. It would have been preferable to conduct flow cytometry on primary cells isolated from IBD or CRC, fluorescently stained for CD31, cytokeratin, vimentin and intracellular IL-34. This would have better illustrated the population(s) of primary cells isolated from IBD or CRC.
Answer: We have taken into account this comment and performed flow cytometry analysis to better illustrated Normal Fibroblasts (NFs) and Cancer Associated Fibroblasts (CAFs) isolated from the same CRC patient as reported in Supplementary Figure S1. NFs and CAFs were negative for CD31 (endothelial marker) and EpCAM (epithelial marker) while they expressed SMA (fibroblast marker). Flow cytometry analysis confirmed the predominant expression of IL-34 in CAFs as compared to NFs.
2. Cell proliferation was analyzed using flow cytometry in figure 4. However, another method to test cell proliferation, e.g., MTT assay, could have been used to use multiple methods to demonstrate IL-34's function.
Answer: Cell proliferation was further assessed by BrdU assay as reported in Supplementary Figure S2. This method confirmed the ability of IL-34 to enhance fibroblast proliferation.
3. The manuscript should better describe and justify the use of the DLD-1 cell line.
Answer: DLD-1 were selected because these cells are known to proliferate vigorously in response to CAF-derived factors (Berdiel-Acer, M.; Neoplasia, 2011: 13: 931-946)

Reviewer 2 Report
The paper investigates the role of IL34 in the tumor-microenvironment interactions of colorectal cancer. The authors use primary fibroblasts isolated from colorectal cancers and adjacent normal colon, which are analysed by standard methodology. In addition, the impact of fibroblast-conditioned media on cancer cells was determined using DLD-1 cells.
The topic is of high interest. However, the work suffers from an essential weakness: the fibroblasts used for the experiments came from only 1 patient each for CRC and IBD. Also only 1 cancer cell line is used to demonstrate the pro-tumorigenic impact of fibroblast-conditioned media. The data from at least 1 more fibroblast population and at least 1 more cancer cell line are necessary to demonstrate the general applicability of the results for CRC pathogenesis.
Author Response
We thank the reviewer for his/her helpful suggestions
We performed experiments using CAFs and NFs isolated from the tumoral area and the macroscopically and microscopically unaffected adjacent non tumoral area of 9 (and not 1) patients who underwent colon resection for sporadic CRC.
Additional surgical samples were taken from tumoral area of 3 patients with colitis-associated colon cancer (CAC).

Round 2
Reviewer 2 Report
I do not really see that the authors addressed my concerns.
With regard to the number of patients analysed, this needs to be clearly detailed in the text and the figure legends.
As for the CRC cell lines: While DLD1 responds very well to stimulation, it is also a cell line representative for the mismatch-defective cancers, not the majority of colonic tumors. Therefore, I still think that a second cell line should be used.
Author Response
We thank the reviewer for his/her helpful suggestions
Answer:
We add the number of patients analyzed in the text and figure legends.
We performed additional experiments and confirmed the fibroblast-derived IL-34 regulatory effects on HT-29 cells as reported in Supplementary Figure 3.
Round 3
Reviewer 2 Report
my concerns were adequately addressed